# Pre-Planned and Non-Planned Agility in Patients Ongoing Rehabilitation after Knee Surgery: Design, Reliability and Validity of the Newly Developed Testing Protocols

**DOI:** 10.3390/diagnostics11010146

**Published:** 2021-01-19

**Authors:** Ivan Peric, Miodrag Spasic, Dario Novak, Sergej Ostojic, Damir Sekulic

**Affiliations:** 1Faculty of Dental Medicine and Health, Josip Juraj Strossmayer University of Osijek, 31000 Osijek, Croatia; ivan.peric@fdmz.hr; 2Faculty of Kinesiology, University of Split, 21000 Split, Croatia; mspasic@kifst.hr (M.S.); dario.novak@inantro.hr (D.N.); 3Institute for Anthropological Research, 10000 Zagreb, Croatia; 4Faculty of Sport and Physical Education, University of Novi Sad, 21000 Novi Sad, Serbia; sergej@panet.rs

**Keywords:** change of direction, reliability, validity, postoperative status, rehabilitation

## Abstract

Background: Due to its association with the risk of falling and consequent injury, the importance of agility is widely recognized, but no study so far has examined the different facets of agility in an untrained/clinical population. The aim of this study was to evaluate the reliability, validity, and correlates of newly developed tests of non-planned agility (NPA) and pre-planned agility (PPA) in an untrained/clinical sample. Methods: The sample comprised 38 participants older than 40 years (22 females, age: 56.1 ± 17.3 years, height: 170.4 ± 10.8 cm, mass: 82.54 ± 14.79 kg) who were involved in a rehabilitation program following total knee arthroplasty and knee arthroscopy. Variables included age, gender, type of surgery, history of fall, anthropometrics/body composition, and newly developed tests of NPA and PPA. Results: The results showed the high inter-testing- (ICC > 0.95, CV < 9%), and intra-testing-reliability (ICC > 0.96, CV < 9) of the newly developed tests. PPA and NPA were found to be valid in differentiation between age groups (>50 yrs. vs. <50 yrs.), and genders, with better performance in younger participants and males. Only NPA differentiated participants according to type of surgery, with better performance in those who had arthroscopic surgery, than those who had total knee arthroplasty. No differences in NPA and PPA were established between groups based on fall-history. In females, the body mass (Pearson’s *r* = 0.58 and 0.59, *p* < 0.001) and body fatness (Pearson’s *r* = 0.64 and 0.66, *p* < 0.001) were negatively correlated, while the lean body mass (Pearson’s *r* = 0.70 and 0.68, *p* < 0.001) was positively correlated with PPA and NPA. The NPA and PPA were highly correlated (Pearson’s *r* = 0.98, *p* < 0.001). Conclusions: We found that the proposed tests are reliable when evaluating agility characteristics in an untrained/clinical population after knee arthroplasty/arthroscopy. Further evaluation of the specific validity of the proposed tests in other specific subsamples is warranted.

## 1. Introduction

Fall-related injuries are a major issue for the human body, particularly among untrained adults and elderly people, with the percentage of injuries constantly rising [1,2,3]. One-third of the population aged 65 or older experiences a fall at least once a year, and this number rises to over half of the population when discussing individuals above the age of 80 experiencing falls at least once per year [4]. About 30% of community-dwelling seniors fall annually and half of these have recurrent falls. This is very important to highlight because 95% of people aged 65 and over are community-dwelling [5]. 

It is well-established that stability requires a well-functioning muscular-skeletal system, as well as an intact balance system [5,6,7]. That being said, agility is considered to be one of the most crucial factors preventing falls and injuries sustained from them in untrained adults [8]. In relation to sport, agility is defined as a rapid, whole-body movement with changes of velocity or direction in response to a stimulus [9]. To adjust this definition to the needs of untrained adults, some authors have defined agility as the quickness of systematic motion [10]. These data underline the need for valid and reliable assessment tools for assessing agility as an important factor influencing the risk of falling and subsequent injuries.

Regarding the previous definition of agility, it is also necessary to explain the difference between two facets of agility: pre-planned agility (PPA) and non-planned agility (NPA). For PPA, the change of direction is planned in advance by an individual. The individual knows when and in what direction they have to perform this movement, and therefore, do not have to respond to an external stimulus [11]. Meanwhile, NPA (also named reactive agility) indicates the ability to change the direction in response to external stimuli [12]. Numerous previous studies have demonstrated that different factors affect the performance of NPA and PPA; however, these were mostly carried out on professional athletes [13,14,15]. Moreover, studies on athletes have indicated that PPA and NPA rarely share more than 20% of the common variance, highlighting the necessity of independent evaluations of these capacities [12,16,17].

Currently, there is a global consensus that agility deserves to be studied outside of the population of professional athletes [5]. This quality indirectly emphasizes the overall motor performance and mobility. Furthermore, it is assumed that agility is directly correlated with the risk of falling in older persons [5,18,19]. The global leader in health-related physical exercising—the American College of Sports Medicine (ACSM)—in their recommendations for health-related physical activity, highlights the necessity of the “development of neuromuscular capacities aimed at improvement of agility, balance, and coordination of minimally 2-to-3 times per week” [8]. However, the first prerequisite of any systematic and reasonable treatment is the reliable and valid measurement of the targeted physical capacity. Not surprisingly, studies have already investigated and evaluated the applicability of various tests aimed at evaluations of agility in untrained subjects. The investigation by Miyamoto and colleagues examined the validity of a new performance test focusing on PPA among community-dwelling and elderly people [10]. In brief, the authors confirmed the criterion-related validity of the Ten-Step Test (TST) as a new performance test for investigating agility among individuals from the community-dwelling group [10]. Manderoos and colleagues presented a new Agility Test for Adults (ATA) and investigated the test-retest reliability [18]. The results showed that ATA is stable and reliable when evaluating agility characteristics in untrained adults. Finally, Sobolewski et al. evaluated the specific test of NPA and suggested its applicability in older persons [20].

From the overview of previous literature, it is evident that, despite the importance of agility in everyday life, only a few studies have examined this problem outside of competitive sports [10,18,20]. Furthermore, to the best of our knowledge, no study has simultaneously examined the reliability and usefulness of real-life NPA and PPA tests among untrained people, which is particularly important knowing the theoretical independence of these capacities [11]. Namely, tests used so far evaluate NPA or PPA, while it would be useful to design a measurement tool that would be useful in evaluation of both capacities in non-athletic samples while also including natural (i.e., regular) movement templates and scenarios. Finally, knowledge about correlates of NPA and PPA in an untrained population is particularly scarce. Therefore, the aim of this study was to evaluate the reliability, validity, and usefulness of newly developed NPA and PPA tests among subjects involved in the rehabilitation process after knee arthroplasty and knee arthroscopy (clinical population). Additionally, we evaluated anthropometric indices, age and gender as factors associated with NPA and PPA in studied participants. Initially, we hypothesized that new real-life NPA and PPA tests would be reliable and valid tests when evaluating agility characteristics in a clinical population.

## 2. Materials and Methods

### 2.1. Participants

Participants in this study were 38 untrained adults, all older than 40 years (56.1 ± 17.3 years). All participants were patients attending a rehabilitation program after arthroscopic surgery and total knee arthroplasty, at the Clinical Medical Centre in Osijek (Osijek, Croatia). The required sample of participants was a-priori calculated on the basis of a pilot study performed on 10 participants of similar fitness status [21]. Specifically, based on descriptive statistics obtained for NPA and PPA tests, statistical power of 0.8, and significance level of 0.05, the required sample of participants was 34.

While we intended to evaluate the applicability, reliability and validity of PPA and NPA tests in untrained/clinical population, the criteria for inclusion of participants in this study were: older than 40 years, no reported complications after arthroscopic surgery or total knee arthroplasty, normal process of rehabilitation following arthroscopic surgery or total knee arthroplasty. Exclusion criteria included inability to walk without knee pain for a distance of 100 m (10 × 10 m), pain while sitting down and standing up from the chair, pain and short of breath during 5 min walking at a normal pace. Further, participants were not included in the investigation if they had balance disorder while standing/walking (irrespective of accompanied wooziness, and/or sensation of movement, spinning, floating), or if they consumed drugs that could be used in treatment of balance disorders (i.e., aminoglycoside antibiotics, Beta-histine). Each participant provided written consent after being informed of the purpose, benefits, and risk of the study. Ethical approval was obtained from the human research ethics committees of the Clinical Medical Centre in Osijek, Croatia and University of Split, Faculty of Kinesiology (Ethical Board Approval No: 2181-205-02-05-18-002) approved on 8 January 2018.

### 2.2. Variables

Variables included age (in years), gender (male-female), anthropometric/body build indices, fall history, PPA, and NPA. Additionally, participants were grouped according to the surgical treatment they had (arthroscopic surgery and total knee arthroplasty).

Anthropometric measures included body height (measured by a Lange anthropometer) and body mass (measured by the OMRON BF511 scale (Kyoto, Japan)). Additionally, the OMRON scale was used to measure body fat (in kg and as a percentage of body mass) and lean body mass (in kg and as a percentage of body mass). Measurement was conducted by an experienced technician, in a closed facility, before the PPA and NPA testing (see later section), between 8:00 and 10:00 a.m.

Fall history was derived from the questionnaire related to fall risk [22]. Participants responded to a questionnaire on the first day of testing, before anthropometric measurement. 

Measurements of the PPA and NPA were performed by a hardware device system based on an ATMEL microcontroller, as the core of the system (model AT89C51RE2; ATMEL Corp, San Jose, CA, USA). A photoelectric infrared sensor (PIS) was used as an external time triggering input (E18-D80NK), and LEDs were used as controlled outputs. The PIS used has been reported to be reliable as a high-speed sensor, with a response time of <2 ms (>500 Hz) and a digital output signal. The sensor’s detection distance ranges from 3 to 80 cm, which made it very applicable for our tests, since participants did not have to stay very close to the sensor. The sensor is connected through a microcontroller IO port. For the purposes of our study, this device was connected to a laptop PC operating on the Linux OS, but other operating systems are also applicable.

The PPA and NPA tests were performed in the testing area shown in Figure 1.

For the NPA, the participants started in the standing position, with their feet parallel to the start line. They then stepped forward with their preferred foot and when crossing the infrared IR signal, the timer began. At that moment, the microcontroller (MC) lit one of three LEDs placed inside 30-cm-high cones, placed below the chairs (labeled 1, 2, and 3). A participant had to assess which cone was lit, walk to that particular chair, sit down, stand up, return to the start line as quickly as possible, turn back, and start over again. When they crossed the IR signal on their way back after the fourth attempt, the timer stopped. Therefore, a single-test trial for each test consisted of four attempts. Participants had no advanced knowledge of the testing scenario, although the testing scenarios for all participants were the same. Three trials were performed with the following scenarios: Trial 1: 1-3-2-3; Trial 2: 2-1-3-2; and Trial 3: 3-1-1-2. The rest period between trials lasted 3 minutes. The PPA testing was conducted at the same testing polygon, but the participant had an advanced knowledge of which cone would light up, and the testing scenario was maximally simplified (Trial 1: 1-2-3-1; Trial 2: 2-3-1-2; and Trial 3: 3-1-2-3). After reliability analyses (see later section for details), the best attempt was used as a final result for both tests. All participants were familiarized with testing procedures throughout 4-5 trials of PPA and NPA a day before testing. In familiarization trials participants performed tests in submaximal tempo and with only 3 attempts for NPA and 2 attempts for PPA using various scenarios. Familiarization trials were not recorded and used for analyses in this study.

### 2.3. Statistics

After assessing the normality by the Kolmogorov–Smirnov test, the means and standard deviations were reported for all variables. The homoscedasticity of all variables was proven by Levene’s test.

Reliability of the tests was examined by calculating inter-testing reliability (test-retest reliability), and intra-testing reliability. A subsample consisting of 13 participants was tested by testing and retesting (two days in a row), in order to establish the inter-session reliability of the PPA and NPA. The intra-session reliability was calculated on the basis of the results of all participants (*n* = 38). The relative reliability was analyzed on the basis of a 2-way mixed model of intraclass correlation coefficient (ICC_3,1_), and the ICC of 0.75 or above are considered to indicate good reliability [23,24]. The absolute reliability was analyzed using the coefficient of variation (CV). Also, standard error of measurement (SEM) was calculated (SEM = square root of the mean square error term from the analysis of variance, with test and retest observed as repeated measures). Finally, the SEM was used to calculate the Minimal Detectable Change (MDC; minimal amount of change that can be interpreted as a real change in measured capacity for an individual with smaller MDC indicating a more sensitive measure), with the following formula: MDC = SEM × 1.96 × √2 [25].

The construct validity of the NPA and PPA tests was evaluated by examining the differences between specific groups of participants based on four grouping-criteria. First, based on results from previous studies where the age of 50 years was indicated as a critical point for a significant decrease in agility, we evidenced differences between two groups of participants, those younger than 50 years, and those older than 50 years [10]. Second, while previous studies regularly confirmed better PPA and NPA in males than in corresponding females (i.e., males and females involved in the same sport, participants of similar status) [17,26], construct validity of PPA and NPA was identified according to gender, by comparing males vs. females. Third, differences in PPA and NPA were evidenced between groups of participants on the basis of the surgical treatment they had (arthroscopic surgery [ART] vs. total knee arthroplasty [TKA]). Since ART is known to be a less demanding surgical procedure than TKA, we hypothesized that the ART group would achieve better results in NPA and PPA. Fourth, we evidenced differences between groups of participants on the basis of their history of fall where we expected better results in patients with no history of fall. In all cases, analysis of variance (ANOVA), with subsequent analysis of covariance (ANCOVA) were applied to identify the differences between specific groups (younger vs. older, males vs. females, ART vs. TKA, fall-history vs. no-history). Specifically, differences were first evaluated by ANOVA. Since age was found as a significant contributing factor of NPA and PPA (please see later Results for details), in the second phase differences in NPA and PPA were additionally analyzed by ANCOVA, while statistically controlling differences in age.

The associations between applied variables were established by Pearson’s correlation coefficients, and the magnitude for coefficients were considered as trivial (<0.1), small (between 0.1 and 0.3), moderate (between 0.3 and 0.5), large (between 0.5 and 0.7), very large (between 0.7 and 0.9), and nearly perfect (>0.9) [27,28].

A level of statistical significance of 95% (*p* < 0.05) was applied. Statsoft’s Statistica ver. 12.0 (Tulsa, OK, USA) was used for all analyses.

## 3. Results

Test-retest reliability for the newly developed PPA and NPA tests is presented in Table 1.

In short, reliability of the PPA and NPA was similar, both with regard to absolute, and relative-reliability. However, the SEM of NPA was somewhat bigger (1.71 and 1.82, for PPA and NPA, respectively), which consequently generated larger MDC in NPA, than in PPA (5.03 and 2.58 for NPA and PPA, respectively). Mean testing results of the 2nd testing day were somewhat better than results obtained on the 1st testing day, but with no significant differences obtained by t-test for dependent samples (*t* value = 0.98 and 0.91, for PPA and NPA, respectively; *p* > 0.05).

Table 2 presents descriptive statistics for all measured variables and intra-testing reliability parameters for PPA and NPA. The intra-testing reliability of the NPA and PPA was high, with similar absolute- and relative-reliability for both measurement tools. 

Figure 2 presents results of the PPA and NPA with differences between age-groups. The ANOVA revealed significant differences in both performances (*F*-test = 5.72 and 4.69, *p* < 0.05), and younger participants (<50 years) achieved better results than the older group (>50 years).

Table 3 presents results of analyses of differences in PPA and NPA calculated between groups based on gender (e.g., males vs. females), type of surgery (e.g., ART vs. TKA), and fall history (fall history vs. no history of fall). ANOVA calculated for PPA and NPA between genders showed significantly better results in males than in females in both tests (*F*-test = 7.07 and 6.05, *p* < 0.05, for PPA and NPA, respectively). In order to control possible influence of age on established gender differences, the ANCOVA with age as covariate was additionally performed, and confirmed better results in males for both tests (*F*-test = 6.81 and 5.70, *p* < 0.05, for PPA and NPA, respectively). 

Differences in NPA and PPA between groups based on surgical treatment calculated by ANOVA showed better achievement in both tests for ART (*F*-test = 11.78 and 10.75, *p* < 0.01, for PPA and NPA, respectively). When controlled for differences in age, ANCOVA evidenced significant differences between groups only for NPA (*F*-test = 4.70, *p* < 0.05), but not for PPA (*F*-test = 4.04, *p* = 0.06). The differences between groups based on history of fall calculated by ANOVA and ANCOVA were not significant. 

Correlations between studied variables for total sample are presented in Table 4. Both agility performances were negatively correlated with body fatness variables. Meanwhile, higher lean body mass positively influenced PPA and NPA. The participants’ age was positively correlated with body fat percentage (moderate-to-large correlations), and negatively correlated with lean body mass (moderate correlation).

A clearer picture of associations was evidenced when correlations were calculated separately for males and females. In females, PPA and NPA were negatively correlated with body mass (large correlation), and body fat (large correlation), and lean body mass positively influenced PPA and NPA (large correlation) (Table 5).

The anthropometrics and body composition did not significantly influence the PPA and NPA in males (Table 6).

## 4. Discussion

The aim of this study was to investigate the reliability, validity, applicability and correlates of newly developed NPA and PPA tests in untrained people. There are several important findings of the study. First, the intra-testing and inter-testing reliability of the developed tests were appropriate and similar for both measurement tools. Second, the newly developed tests were found to be discriminative according to subjects’ age and gender. Third, NPA significantly differentiated groups according to surgical procedure, but no differences were found between groups based on fall history. Therefore, our initial study hypothesis may be partially accepted. Finally, the NPA performance was negatively correlated with participants’ age for both genders, while PPA was correlated with anthropometric/body build indices in women. 

### 4.1. Reliability of the Newly Developed PPA and NPA in Untrained/Clinical Subjects

Our study evidenced the high reliability of NPA and PPA tests, obtaining a high relative reliability and appropriate absolute reliability. The similar reliability of both tests is somewhat surprising, since previous studies on professional and semi-professional athletes consistently reported a lower reliability for the NPA protocols in comparison to corresponding PPA protocols (note that “corresponding” denotes protocols consisting of similar movement scenarios, distances, etc., with differences in the “reactive/non-planned” vs. “non-reactive/pre-planned” nature of the tests). In short, a study on basketball players showed a better reliability for the basketball specific PPA in comparison to the basketball specific NPA tests (ICC of 0.81 and 0.87 for NPA and PPA, respectively) [15]. Moreover, the reliability of the soccer specific PPA was better than the corresponding NPA test in youth soccer players (ICC of 0.88 and 0.92 for NPA and PPA, respectively) [14], while similar conclusions were also presented in handball and futsal studies as well [13,17]. In explaining the relative inconsistency between our results (i.e., similar reliability of PPA and NPA in our participants) and those presented in previous studies (i.e., better reliability of PPA than NPA tests in previous athletic studies), the descriptive statistics should be overviewed.

In brief, our subjects achieved similar results in PPA and NPA, meaning there was no significant difference between studied capacities (see Results for more details). On the other hand, practically all previous investigations reported a 10–20% better performance for PPA in comparison to corresponding NPA tests [13,15,17]. Although this may seem surprising, the similar achievement in NPA and PPA among our subjects is explainable knowing the specifics of the proposed testing protocols. Namely, in this study, PPA and NPA tests were performed by walking. On the other hand, studies conducted with professional athletes comprised the maximal running performance (i.e., sprinting) during PPA and NPA testing, simply because, in a competitive environment (i.e., games), PPA and NPA are performed as rapidly as possible [13,15,17]. In such circumstances (maximal effort and arousal in running), the cognitive processing and perceptual capacities are challenged, the anticipation of the visual signal is altered, and the reaction time is slower [29]. Since all of these factors (anticipation, cognitive processing, and the reaction time) directly affect the NPA performance [30], the approach naturally resulted in relatively slower NPA in running tasks [15]. Meanwhile, when NPA and PPA are performed while walking, as in our study, the perceptual/cognitive capacities important for NPA are probably not so burdened. This consequently reduces the negative influence of necessary cognitive processing in execution of the NPA, altogether resulting in the similar NPA and PPA performances obtained in our study. 

### 4.2. Validity of the Newly Developed PPA and NPA Tests in Untrained/Clinical Subjects

So far, several tests have been used as a part of test batteries for evaluating the agility components in untrained people [4,10,18]. For example, Manderoos and colleagues (2015) presented a new Agility Test for Adults [18]. Miyamoto et al. (2008) evaluated a new performance test focusing on agility among community-dwelling and elderly people and Dite et al. (2002) constructed a new clinical test of the dynamic static balance [4,10]. All of these tests were useful for evaluating the agility performance of adult participants. However, our results provide additional information on the problem. Namely, the simultaneous investigation of the PPA and NPA performances conducted in our study allowed us to compare the construct validity of the developed tests in untrained people.

First, we must note that NPA and PPA were strongly correlated, and the correlation between these two performances indicated that tests share more than 90% of the common variance. This is a relatively unique finding, since previous studies regularly reported independence of these performances, with common variance rarely exceeding 20% [12,16,17]. However, to the best of our knowledge this is the first study where different facets of agility (e.g., pre-planned and non-planned agility) were simultaneously examined in the clinical population. On the other hand, studies done with athletes (e.g., trained individuals) indicated that PPA primary depends on conditioning capacities (i.e., speed, strengths, power, balance), the NPA-performance is highly influenced by perceptual, cognitive and reactive capacities of the tested participants [30]. However, all studies where authors confirmed relative independence of the PPA and NPA (e.g., low correlation between PPA and NPA) included well-trained athletes, whose results on tested capacities were not variable [13,14,30]. Meanwhile, the performances of our participants were highly variable (i.e., some participants performed 3-times better than others). Simply mathematically, such high variability of the results increased the correlation between NPA and PPA in the observed sample [31]. 

Both NPA and PPA were found to be valid in the differentiation of the age groups observed, and in both cases, the younger group (<50 years of age) achieved better results than the older group (>50 years of age). Although we could not find a study that simultaneously investigated NPA and PPA in an untrained population, knowing the aging-related neuromuscular and neurocognitive declines, we may say that evidenced differences were expected [32]. Namely, studies have consistently reported a negative correlation between age and various indices of the fitness status in untrained adults. Such results were found for the handgrip strength [33], aerobic capacity [34], and flexibility [35]. Moreover, our results confirm the findings of a Japanese study, where Miyamoto et al. highlighted a decline in agility after 50 years of age [10].

Males achieved significantly better results than females in NPA and PP even when differences were controlled for age by analysis of covariance. There are several plausible explanations of such findings. First, there is a possibility that male participants were generally in better fitness than females, which is regularly accentuated in studies where agility performances were compared across genders [17,26]. In our study this is particularly possible knowing the age of our participants (>40 years), and negative influence of metabolic and physiological changes associated with menopause on overall fitness status of females [36]. We must note that this is particularly possible since body fat was evidenced as a significant factor of influence on NPA and PPA in females (please see later for discussion). As a result, our tests can be observed as valid tools in differentiation of agility facets between genders for clinical population.

Although the ART group achieved better results than TKA group in both agility performances, ANCOVA controlled for age evidenced significant differences only for NPA. Therefore, irrespective of high correlation between PPA and NPA and previously discussed similar construct validity of both measurement tools with regard to differentiation between genders and age-groups, only NPA was found to be valid for the purpose of differentiation between groups based on surgical treatment (e.g., ART and TKA). Knowing the differences between these two surgical treatments we may say that our results are expected. Also, this finding deserves specific attention as it highlights the necessity of independent evaluation of NPA and PPA in the clinical population. The background of the construct validity established exclusively for NPA is almost certainly related to differences in locomotor efficacy and musculoskeletal functionality (functionality of the knee joint) between ART and TKA.

With regard to functionality of the knee, differences between surgical treatments are of utmost importance. Namely, while ART surgical treatment preserves the natural knee joint, the TKA operation actually replaces the knee joint with an artificial joint. As a result, the knee proprioception, and balance capacity are better preserved after ART [37,38,39]. Another important difference between ART and TKA surgery is the size of the incisions. In ART, three tiny incisions (about 1 cm) are made in the knee joint. On the other hand, TKA is a so-called “open procedure”, with a relatively long incision. As a result, lower functionality of knee-joint muscles is expected after TKA. It altogether may explain even the differences between ART and TKA in agility tests observed here.

The proprioceptive capacities of the joints are more challenged in NPA. This is mainly a result of the fact that during the NPA participants are not able to “pre-program” the movement template but had to react to unpredictable stimuli [30]. This logically led to better performance in NPA among the ART group simply because of the superior functionality of the knee (due to previously explained differences between surgeries). Of course, differences in functionality of the musculoskeletal system could influence the PPA capacity as well and this is supported even in ANOVA results where ART achieved better results than TKA. However, age confounded the results of PPA significantly, resulting in significant ANCOVA differences between ART and TKA only for NPA. Collectively, we may say that our study confirmed proper construct validity of the NPA in evaluation of the differences between observed surgical treatments.

To the best of our knowledge this is the first study where history of fall is observed as a factor associated with agility performances in untrained/clinical samples. Knowing the association between balance performance with risk of fall [40], as well as positive correlation between balance and agility [26], we hypothesized impaired NPA and PPA in participants who experienced fall. However, our results did not confirm such hypothesis. Mainly, it seems that differences in age between groups led to such results at least partially. Specifically, additional analyses allowed us to identify that participants who experienced fall were significantly younger than those with no fall history (43.72 ± 21.47 vs. 59.96 ± 14.24, respectively; *F*-test = 6.96, *p* = 0.01). While age is found to be the most significant factor of influence on NPA and PPA it logically limited the possibility to evidence significant differences between groups based on fall-history. Therefore, for the moment we are not able to highlight appropriate construct validity of the NPA and PPA in differentiating participants who experienced fall from those with no history of fall. However, further studies are needed with more age-balanced groups in order to evaluate validity of the proposed tests for distinguishing participants according to their fall-status.

### 4.3. Correlates of the Newly Developed NPA and PPA Performances in Untrained/Clinical Subjects

The factors associated with PPA and NPA varied for the total sample, and across genders. In brief, while age negatively influenced agility performances in both genders, the body mass and indices of body build were correlated with NPA and PPA only among females. While the negative influence of age on the agility performance in both genders was previously discussed, in the following text, we will focus on correlations between body mass and body build indices, and agility performances among females. 

Excessive body fat negatively influences the muscular performances by various physiological and metabolic mechanisms (i.e., higher level of body fat increases the muscle temperature and consequently negatively influences the muscular capacities, and excess body fat decreases the plasma fatty acid availability) [41,42]. However, we believe that, in our study, the relationship between body mass and body composition, and agility in women should mainly be contextualized through two mechanisms. The first mechanism is related to the negative influence of body fat measures on the agility performance. Second, general fitness status is characteristic for women with higher body mass and body fat values.

Studies have regularly demonstrated the negative influence of body fat on locomotor performances where the whole body weight (or a significant proportion of the body weight) should be displaced and/or conquered. Although such findings have mostly been reported in competitive sport settings, similar correlations have been found in non-athletic and clinical populations [43,44]. Irrespective of the study population, the explanations are mostly oriented toward body fat as “ballast mass” in performances where efficient body displacement is an important determinant of performance, as in the agility performances [45]. In our study, the negative influence of body fatness and body mass on the studied performances was additionally aggravated by the fact that PPA and NPA include several changes of direction, and in every single change of direction, inertia (resistance of the body to change in its motion) appears. This accentuates the negative influence of body fatness on NPA and PPA as well.

Nevertheless, it is also possible that higher body mass and fat values and a lower lean body mass are specific for females who have a lower level of overall physical fitness. Indeed, while body build indices could be observed as predictors of fitness (i.e., subjects with more developed muscles are stronger, faster, and more agile), it is also possible that physical fitness is a predictor of body build (i.e., subjects with a low fitness level avoid physical activity and therefore become overweight). The latter is particularly possible in adult females, since lifelong metabolic changes (i.e., menopause) often result in detrimental changes in their fitness status, and consequently alienate even formerly active and fit women from physical activity, affecting their body composition [46]. While the precise identification of exact mechanisms was not within the scope of this study, this should be studied in more detail in future investigations.

### 4.4. Methodological Considerations

Before presenting the conclusion, we would like to acknowledge the change in testing protocols during the experiment. Initially, the pilot study was performed, where different forms of NPA and PPA testing procedures were evaluated [21]. In brief, participants originally had to perform a test that was longer in duration (five attempts), with a shorter additional test being added to monitor changes in the agility performance. Commonly used agility tests are generally short in duration and require very few changes of direction [47]. The issue with longer agility tests is the effect that fatigue may have on the performance due to the extended test duration [48]. Despite this, fatigue has been shown to affect the performance and biomechanics in untrained people [49]. We realized that the initial longer protocol had extended the test duration and therefore, a test with a shorter duration needed to be an option that was tested. The test with the shorter duration showed a decrease in fatigue issues. Moreover, the use of a shorter test allowed us to measure the subjects’ real agility performance.

### 4.5. Limitations and Strengths

There are a number of limitations of this study that need to be taken into consideration. First, due to the cross-sectional design, the possibility of reverse causation cannot be excluded, and we have tried to highlight it in a previous text. Furthermore, poor health could have resulted in a low level of agility and other reactive indicators within the agility test design. This may lead to high variability of the results, which consequently increased even the correlation between PPA and NPA in our participants. However, the high heterogeneity of the results is a natural characteristic of the studied sample, and therefore can present the “real nature” of the studied problem. Next, we did not have the possibility to look at the mental and physical fatigue that may have occurred during the testing. This means it could potentially be an aspect that affects the proper execution of the movement. In future studies this issue may be solved by evaluating tests consisting of only one attempt, either for NPA or PPA performance. One could argue that such analysis is possible even now (i.e., every attempt is recorded), but in this study participants were initially informed about the number of attempts and therefore almost certainly did not execute the first attempt maximally. Also, a relatively small number of participants were included in each test group. Therefore, future studies are required to assess real-life agility by approaching more subjects, which will allow for larger groups of test subjects. Finally, this study evaluated only a small number of correlates of PPA and NPA and further studies should evaluate other variables than those studies in this investigation as being potentially correlated to PPA and NPA in the clinical population.

This study was one of the first where NPA and PPA protocols in untrained individuals were simultaneously evaluated. It allowed us to evaluate not only the reliability and applicability of the tests, but also the construct validity, by an examination of its mutual association. Additionally, this is one of the first studies where the sample comprised untrained adults from clinically important populations (i.e., populations where agility components are important determinants of the fitness status). Finally, this study extends the previous knowledge on correlates of NPA and PPA in untrained subjects. Therefore, although not being the final word on the topic, we believe that the study will initiate further research within the field.

## 5. Conclusions

The proposed tests of NPA and PPA were found to be reliable in an evaluation of agility in untrained/clinical subjects. However, the tests were highly intercorrelated, which means that PPA and NPA in non-trained subjects should not be considered as independent qualities. As a result, we may suggest the application of the proposed PPA test as a sufficient measurement tool in evaluation of the agility performance among subjects similar to those here studied. However, while there is a possibility that fatigue influenced testing results, in future studies proposed tests should be modified (i.e., shortened in duration), and checked for association between NPA and PPA performances.

Collectively, we may say that NPA and PPA were equally applicable in identifying the differences between age groups and genders, which indicates similar construct validity of both testing procedures. However, knowing the specifics and differences in the measurement of NPA and PPA, the similar applicability/validity of both tests is particularly important. In brief, the NPA measurement asks for specifically designed equipment while testing of the PPA can be performed by any commercially available and relatively cheap photo cells. Therefore, based on our results, it seems that NPA and PPA should be considered as equally valid in the evaluation of agility performances in untrained adults when it comes to differentiation according to age, and gender.

However, only NPA differentiated participants according to surgery they had. Results are easily connectable to better functionality of knee joint in ART than in the TKA group, and higher importance of proprioceptive neuromuscular capacities in NPA. Therefore, further development and evaluation of the NPA tests should be contextualized with regard to specifics of different surgical treatments and phase of rehabilitation. Most specifically, we are of the opinion that PPA is more important in earlier phases of the rehabilitation process and/or for less demanding operations, while NPA should be tested in later stages of rehabilitation process.

In females, agility was strongly negatively influenced by body mass and body fatness. Therefore, we may suggest the application of agility training in overweight females, in order to reduce the risk of falling and injury.

Further studies are needed to identify the applicability and correlates of the proposed tests in evaluations of agility components in other samples of subjects. More studies assessing the longitudinal validity of the proposed protocols when measuring changes over time are also warranted.

## Figures and Tables

**Figure 1 diagnostics-11-00146-f001:**
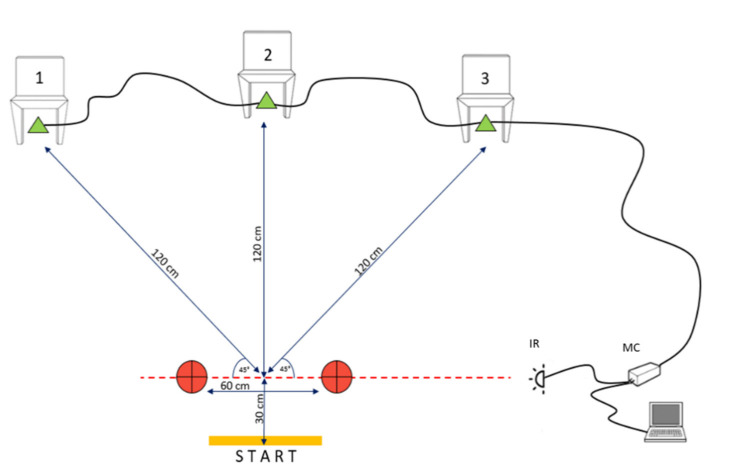
Testing of the pre-planned agility and non-planned agility (IR—infrared sensor; MC—microcontroller).

**Figure 2 diagnostics-11-00146-f002:**
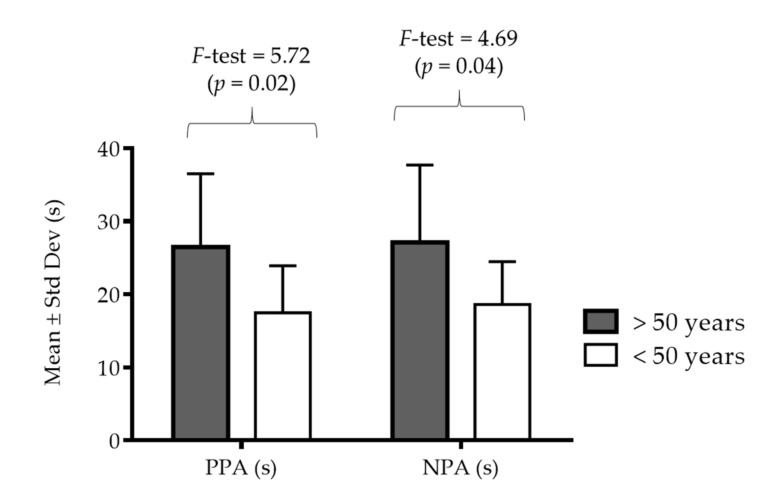
Descriptive statistics for tests of non-planned agility (NPA), and pre-planned agility (PPA) with differences between age groups.

**Table 1 diagnostics-11-00146-t001:** Inter-testing reliability of the newly developed tests of the pre-planned agility and non-planned agility.

	1st Testing Day		2nd Testing Day		Inter-Session Reliability			
	Mean (95%CI)	SD	Mean (95% CI)	SD	ICC (95%CI)	CV	SEM	MDC
PPA (s)	24.09 (22.5–25.6)	2.88	22.81 (21.29–24.32)	2.79	0.97 (0.82–0.99)	0.08	1.71	2.58
NPA (s)	24.08 (22.12–26.03)	3.59	22.29 (20.18–24.39)	3.87	0.95 (0.90–0.99)	0.08	1.82	5.03

Legend: SD—standard deviation, CI—confidence interval, ICC—intra class correlation coefficient, CV—coefficient of variation, SEM—standard error of measurement, MDC—minimal detectable change, PPA—pre-planned agility, NPA—non-planned agility.

**Table 2 diagnostics-11-00146-t002:** Descriptive statistics for the measured variables and intra-testing reliability of the pre-planned and non-planned agility.

	Mean	Min	Max	SD	ICC (95% CI)	CV	SEM	MDC
PPA (s)	24.58	10.82	49.95	10.02	0.97 (0.90–0.99)	0.04	1.56	4.28
NPA (s)	25.33	11.9	57.5	10.35	0.97 (0.89–0.99)	0.05	1.60	4.39
Body height (cm)	170.42	152.6	193.7	10.82				
Body mass (kg)	82.54	55.5	124.1	14.79				
Body fat (kg)	26.35	6.44	49.76	10.46				
Body fat (%)	31.62	11.6	51.9	10.46				
Lean body mass (kg)	24.83	15.93	38.5	5.83				
Lean body mass (%)	30.27	19.7	41.3	5.58				

Legend: SD—standard deviation, CI—confidence interval, ICC—intra class correlation coefficient, CV—coefficient of variation, SEM—standard error of measurement, MDC—minimal detectable change, PPA—pre-planned agility, NPA—non-planned agility.

**Table 3 diagnostics-11-00146-t003:** Analysis of the differences in agility performances between groups based on gender, type of surgery and fall history.

	Mean ± SD	Mean ± SD	ANOVA	ANCOVA
			*F* TEST(*p*)	*F* TEST(*p*)
Gender	Males	Females		
PPA (s)	19.88 ± 6.42	27.99 ± 10.88	7.07(0.01)	6.81(0.01)
NPA (s)	20.79 ± 5.93	28.63 ± 11.68	6.05(0.02)	5.70(0.02)
Type of surgery	Arthroscopic surgery	Total knee arthroplasty		
PPA (s)	19.37 ± 6.27	29.27 ± 10.56	11.98(0.001)	4.7(0.037)
NPA (s)	20.17 ± 5.69	29.98 ± 11.47	10.75(0.002)	4.04(0.06)
Fall history	Yes	No		
PPA (s)	19.8 ± 6.96	26.06 ± 10.45	2.81(0.10)	0.530(0.47)
NPA (s)	21.09 ± 6.33	26.65 ± 11.07	2.04(0.16)	0.228(0.64)

Legend: SD—standard deviation, ANOVA—analysis of the variance, ANCOVA—analysis of the covariance with age as covariate, PPA—pre-planned agility, NPA—non-planned agility.

**Table 4 diagnostics-11-00146-t004:** Pearson’s correlation coefficients between studied variables for total sample of subjects.

Total (*n* = 38)	PPA (s)	NPA (s)	Age (Years)	Body Height (cm)	Body Mass (kg)	Body Fat (kg)	Body Fat (%)	Lean Body Mass (kg)
NPA (s)	0.98 ***							
Age (years)	0.42 **	0.41 **						
Body height (cm)	−0.33 *	−0.28	−0.45 **					
Body mass (kg)	0.19	0.18	0.08	0.36 *				
Body fat (kg)	0.52 ***	0.48 **	0.30	−0.43 *	0.57 ***			
Body fat (%)	0.56 ***	0.52 ***	0.34 *	−0.70 ***	0.16	0.89 ***		
Lean body mass (kg)	−0.24	−0.21	−0.30	0.86 ***	0.59 ***	−0.29	−0.63 ***	
Lean body mass (%)	−0.54 ***	−0.48 **	−0.46 **	0.72 ***	−0.19	0.88 ***	0.96 ***	0.66 ***

Legend: PPA—pre-planned agility, NPA—non-planned agility, *** *p* < 0.001, ** *p* < 0.01, * *p* < 0.05.

**Table 5 diagnostics-11-00146-t005:** Pearson’s correlation coefficients between studied variables for female subjects.

Females (*n* = 22)	PPA (s)	NPA (s)	Age (Years)	Body Height (cm)	Body Mass (kg)	Body Fat (kg)	Body Fat (%)	Lean Body Mass (kg)
NPA (s)	0.98 ***							
Age (years)	0.41	0.42 *						
Body height (cm)	−0.13	−0.08	−0.39					
Body mass (kg)	0.68 ***	0.67 ***	0.14	0.28				
Body fat (kg)	0.70 ***	0.64 ***	0.35	−0.41	0.66 ***			
Body fat (%)	0.57 **	0.51 *	0.41	−0.62 **	0.41	0.94 ***		
Lean body mass (kg)	0.08	0.12	−0.26	0.80 ***	0.54 **	−0.26	−0.54 **	
Lean body mass (%)	−0.55 **	−0.49 *	−0.46 *	0.63 **	−0.33	−0.90 ***	−0.99 ***	0.60 **

Legend: PPA—pre-planned agility, NPA—non-planned agility, *** *p* < 0.001, ** *p* < 0.01, * *p* < 0.05.

**Table 6 diagnostics-11-00146-t006:** Pearson’s correlation coefficients between studied variables for male subjects.

Males (*n* = 16)	PPA (s)	NPA (s)	Age (Years)	Body Height (cm)	Body Mass (kg)	Body Fat (kg)	Body Fat (%)	Lean Body Mass (kg)
NPA (s)	0.98 ***							
Age (years)	0.46	0.42						
Body height (cm)	−0.11	−0.09	−0.67 **					
Body mass (kg)	0.2	0.09	0.16	−0.12				
Body fat (kg)	0.12	0.04	0.22	−0.43	0.88 ***			
Body fat (%)	0.09	0.04	0.21	−0.54 *	0.74 ***	0.96 ***		
Lean body mass (kg)	0.02	0.01	−0.45	0.75 ***	0.22	−0.19	−0.34	
Lean body mass (%)	−0.21	−0.13	−0.48	0.65 **	−0.74 ***	−0.93 ***	−0.92 ***	0.47

Legend: PPA—pre-planned agility, NPA—non-planned agility, *** *p* < 0.001, ** *p* < 0.01, * *p* < 0.05.

## Data Availability

The data presented in this study are available on request from the corresponding author. The data are not publicly available due to founder’s policy.

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
