# Peer review of "Pre-Planned and Non-Planned Agility in Patients Ongoing Rehabilitation after Knee Surgery: Design, Reliability and Validity of the Newly Developed Testing Protocols"

_diagnostics, 2021, doi:10.3390/diagnostics11010146_

Round 1
Reviewer 1 Report
The authors have met all the revisions and the proof looks good. Great work by the authors making adjustments to make this paper publishable.
Author Response
The authors have met all the revisions and the proof looks good. Great work by the authors making adjustments to make this paper publishable.
RESPONSE: Thank you for recognizing the quality of our work.
Reviewer 2 Report
Minor comments
„This data underlines the need for valid and reliable assessment tool for assessing agility as important factor influencing the risk of falling and subsequent injuries.” - I recommend to move this paragraph after introducing the term agility.
Author Response
„This data underlines the need for valid and reliable assessment tool for assessing agility as important factor influencing the risk of falling and subsequent injuries.” - I recommend to move this paragraph after introducing the term agility.
RESPONSE: Thank you for your suggestion. Text is amended accordingly, and the mentioned statement is placed at the end of the 2nd paragraph of the Introduction, which now reads: "It is well-established that stability requires a well-functioning muscular-skeletal system, as well as an intact balance system [5-7]. That being said, agility is considered to be one of the most crucial factors preventing falls and injuries sustained from them in untrained adults [8]. In relation to sport, agility is defined as a rapid, whole-body movement with changes of velocity or direction in response to a stimulus [9]. To adjust this definition to the needs of untrained adults, some authors have defined agility as the quickness of systematic motion [10]. This data underlines the need for valid and reliable assessment tool for assessing agility as important factor influencing the risk of falling and subsequent injuries."
Staying at your disposal.
Reviewer 3 Report
The extensive revision of the text and of list of references is fine. I have no further suggestions to improve this typescript
Author Response
The extensive revision of the text and of list of references is fine. I have no further suggestions to improve this typescript
RESPONSE: Thank you for recognizing the quality of our work.
This manuscript is a resubmission of an earlier submission. The following is a list of the peer review reports and author responses from that submission.
Round 1
Reviewer 1 Report
The main objective of this paper is to propose two new tests to assess agility in clinical population. The subject is of interest, but there are very important methodological weaknesses, that cannot be solved, which do not counsel its publication in the present form.
Specific concerns:
- The Title is inaccurate. A more specific approach is necessary, such as: Design, reliability and concurrent validity of a pre-planned and a non-planned agility tests in patients ongoing rehabilitation after knee surgery.
- The Abstract should be adapted to the new comments and modifications of the text.
- Line 44: The need for valid and reliable assessment tools for assessing agility should be included in this sentence: “This data underlines the need for interventions that will improve stability and agility ….”
- Line 85: Please, explain in the Introduction why usefulness is necessarily a part of your aims.
What do the authors mean with: certain correlates? In light of the Results section, many other variables could increase the relevance of the results, such as balance, quality of life, pain,…
- Line 96: The exclusion criteria should be better detailed. Did not the authors exclude vestibular diseases? And previous surgeries in the low limbs? And what about drugs?
- Line 141 (Statistics): This section is poorly detailed. Each aim should be associate to a specific statistical analysis, and the thresholds to accept each hypothesis were not explained and justified.
An a priori sample size calculation is lacking. Anyway, a power analysis of the results is also lacking.
Ten participants for inter-session reliability could be insufficient. It is hard to assume that all variables were normally distributed independent of the group and subgroups sizes.
Which type of intraclass correlation coefficient was applied? How was it interpreted? And the Pearson coefficient?
Other absolute reliability parameters, such as Standard error of measurement or Minimum detectable change, should be reported.
- Results: For a better description of the sample and the improvement of external validity of the study design, please, report the sociodemographic and clinical variables of the participants. Relevant data, such as pain, time after the surgery,… are relevant.
- The division by age is interesting, but why 50 years? And other subdivisions according to different factors were also studied?
- Tables 1 and 2: The 95% confidence intervals are necessary for all outcomes.
- Tables 3 and 4: Again, why anthropometric variables were the only to be compared with outcomes?
- All Discussion and Conclusion sections should be adapted to all new perspectives.
The high correlation between both tests results could be due to a fail in construct validity. A deep discussion of this possibility could improve some parts of the text.
The heterogeneity of the sample needs further and focused comments.
- Line 328: A proof concept could solve these problems in the design of a new test for physical features.
- Please, revise the references, because some details are lacking (for example: 2, 10, 23).
Author Response
Reviewer 1
The main objective of this paper is to propose two new tests to assess agility in clinical population. The subject is of interest, but there are very important methodological weaknesses, that cannot be solved, which do not counsel its publication in the present form.
RESPONSE: Thank you for your comments and suggestions. We have tried to follow it specifically, an amended the manuscript accordingly. Please see below for responses on each of your comments.
Specific concerns:
The Title is inaccurate. A more specific approach is necessary, such as: Design, reliability and concurrent validity of a pre-planned and a non-planned agility tests in patients ongoing rehabilitation after knee surgery.
RESPONSE: Thank you for your accurate suggestion. The title is changes and now reads: Design, reliability and discriminative validity of a pre-planned and a non-planned agility tests in patients ongoing rehabilitation after knee surgery.
- The Abstract should be adapted to the new comments and modifications of the text.
RESPONSE: The abstract is systematically rewritten according to modifications done in the text. Thank you.
- Line 44: The need for valid and reliable assessment tools for assessing agility should be included in this sentence: “This data underlines the need for interventions that will improve stability and agility ….”
RESPONSE: Thank you for your suggestion. The sentence is rewritten and now reads: “This data underlines the need for valid and reliable assessment tool for assessing agility as important factor influencing the risk of falling and subsequent injuries” (please see end of first paragraph of the Introduction)
- Line 85: Please, explain in the Introduction why usefulness is necessarily a part of your aims.
RESPONSE: In this version of the manuscript we tried to emphasize the “logics” of the “usefulness” and text reads: “Namely, tests used so far evaluate wither NPA or PPA, while it would be useful to design the measurement tool which will be useful in evaluation of both capacities in non-athletic sample while including natural (i.e. regular) movement templates and scenarios.” (please see highlighted text in the last paragraph of the Introduction).
What do the authors mean with: certain correlates? In light of the Results section, many other variables could increase the relevance of the results, such as balance, quality of life, pain,…
RESPONSE: Indeed, the studied correlates were specified explained in the original version of the paper. Therefore, the part of the text where aims of the study is presented is rewritten and now reads: “Therefore, the aim of this study was to evaluate the reliability, validity, and usefulness of newly developed NPA and PPA tests among subjects involved in the rehabilitation process after knee arthroplasty and knee arthroscopy (clinical population). Additionally, we evaluated anthropometric indices, age and gender as factors associated with NPA and PPA in studies participants.” (please see end of the last paragraph of the Introduction section). Also, we totally agree with your opinion that other variables deserve attention too, and this is now specifically stated in the Limitations of the study. Text reads: Finally, this study evaluated only small number of correlates of PPA and NPA Also, further studies should evaluate other variables than those studies in this investigation as being potentially correlated to PPA and NPA in clinical population.” (please see 1st paragraph of the Limitations and strengths subheading – before Conclusion)
- Line 96: The exclusion criteria should be better detailed. Did not the authors exclude vestibular diseases? And previous surgeries in the low limbs? And what about drugs?
RESPONSE: Thank you for your suggestion. In the revised version of the manuscript more details are added about exclusion criteria. Text reads: Exclusion criteria included inability to walk without knee pain on a distance of 100 meters (10x10 meters), pain while sitting down and standing up from the chair, pain and short of breath during 5 min walk in normal pace. Further, participants were not included in the investigation if they had balance disorder while standing/walking (irrespective of accompanied wooziness, and/or sensation of movement, spinning, floating), or if they consumed drugs that could be used in treatment of balance disorders (i.e. aminoglycoside antibiotics, Beta-histine).” (please see subsection on Participants).
- Line 141 (Statistics): This section is poorly detailed. Each aim should be associate to a specific statistical analysis, and the thresholds to accept each hypothesis were not explained and justified.
RESPONSE: In the revised version of the manuscript we paid special attention to this issue and each study aim was connected to specific statistical analyses. Also, significant changes in analyses are done in accordance with your other comments (i.e. calculation of SEM and MDC; additional analysis of the validity). Therefore, the Statistics subsection is systematically rewritten and now reads:
“After assessing the normality by the Kolmogorov–Smirnov test, the means and standard deviations were reported for all variables. The homoscedasticity of all variables was proven by Levene’s test.
Reliability of the tests was examined by calculating inter-testing reliability (test-retest reliability), and intra-testing reliability. A subsample consisting of 13 participants was tested by testing and retesting (two days in a row), in order to establish the inter-session reliability of the PPA and NPA. The intra-session reliability was calculated on the basis of the results of all participants (n = 38). The relative reliability was analyzed on the basis of 2-way mixed model of intraclass correlation coefficient (ICC3,1), and the ICC of 0.75 or above are considered to indicate good reliability (Green, Blake, & Caulfield, 2011; Shrout & Fleiss, 1979). The absolute reliability was analyzed using the coefficient of variation (CV). Also, standard error of measurement (SEM) was calculated (SEM = square root of the mean square error term from the analysis of variance, with test and retest observed as repeated measures). Finally, the SEM was used to calculate the Minimal Detectable Change (MDC; minimal amount of change that can be interpreted as a real change in measured capacity for an individual with smaller MDC indicating a more sensitive measure), with the following formula: MDC = SEM x 1.96 x √2 (Weir, 2005).
Discriminative validity of the tests was evaluated by examining the differences between specific groups of participants, namely (i) between specific age-groups, and (ii) between genders. In brief, since previous studies showed a significant decrease in agility after 50 years of age (Miyamoto et al., 2008), we identified the differences in NPA and PPA by comparing participants younger than 50 years and those older of 50 years. Second, while previous studies regularly confirmed better PPA and NPA in males than in corresponding females (i.e. males and females involved in same sport, participants of similar status) (Sekulic, Spasic, Mirkov, Cavar, & Sattler, 2013; Spasic et al., 2015), discriminative validity of PPA and NPA was identified according to gender, by comparing males vs. females. In both cases, Student’s t-tests for independent samples was calculated in order to compare groups in PPA and NPA.
Additionally, the differences between PPA and NPA were evidenced by a t-test for dependent samples. The relationships between applied variables were established by Pearson’s correlation coefficients, and the magnitude for coefficients were considered as trivial (< 0.1), small (between 0.1 and 0.3), moderate (between 0.3 and 0.5), large (between 0.5 and 0.7), very large (between 0.7 and 0.9), and nearly perfect (> 0.9) (Hopkins, 2000; Metikos, Mikulic, Sarabon, & Markovic, 2015).
A level of statistical significance of 95% (p < 0.05) was applied. Statsoft’s Statistica ver. 12.0 (Tulsa, OK, USA) was used for all analyses.”
Please see subsection on Statistics. Thank you.
An a priori sample size calculation is lacking. Anyway, a power analysis of the results is also lacking.
RESPONSE: Indeed, in the original version of the manuscript we missed to report that the required sample was calculated on the basis of preliminary pilot study. It is now specified, and text reads: On the basis of results from preliminary pilot study (Perić, Spasić, & Sekulić, 2019), statistical power of 0.8, and significance level of 0.05, the required sample of participants was 34.” (Please see last sentence of Participants subsection)
Ten participants for inter-session reliability could be insufficient. It is hard to assume that all variables were normally distributed independent of the group and subgroups sizes.
RESPONSE: Indeed, the sample size in this case may be small, but the problem is that we studied highly specific sample of participants, and therefore, it was difficult to collect more participants than we did originally. However, following your suggestion, during the revision of the paper we tested 3 additional participants, and in this version of the manuscript we reported the inter-testing reliability for 13 patients. Also, and as you suggested in other comment, the reliability analysis no includes calculation of SEM and MDC. Results are presented in Table 1 and text which reads: “Test-retest reliability for the newly developed PPA and NPA tests is presented in Table 1. In short, reliability of the PPA and NPA was similar, both with regard to absolute, and relative-reliability. However, the SEM of NPA was somewhat bigger (1.71 and 1.82, for PPA and NPA, respectively), which consequently generated larger MDC in NPA, than in PPA (5.03 and 2.58 for NPA and PPA, respectively). Mean testing results of the 2nd testing day were somewhat better than results obtained at 1st testing day, but with no significant differences obtained by t-test for dependent samples (t value = 0.98 and 0.91, for PPA and NPA, respectively; p > 0.05).” (Please see text bellow Table 1, thank you.
Which type of intraclass correlation coefficient was applied? How was it interpreted? And the Pearson coefficient?
RESPONSE: In this version of the manuscript we provided more details about calculation and interpretation of the used statistical parameters. For ICC text now reads: “The relative reliability was analyzed on the basis of 2-way mixed model of intraclass correlation coefficient (ICC3,1), and the ICC of 0.75 or above are considered to indicate good reliability (Green, Blake, & Caulfield, 2011; Shrout & Fleiss, 1979).” (please see 2nd paragraph of the Statistics subsection). For Pearson’s r text reads: “The relationships between applied variables were established by Pearson’s correlation coefficients, and the magnitude for coefficients were considered as trivial (< 0.1), small (between 0.1 and 0.3), moderate (between 0.3 and 0.5), large (between 0.5 and 0.7), very large (between 0.7 and 0.9), and nearly perfect (> 0.9) (Hopkins, 2000; Metikos, Mikulic, Sarabon, & Markovic, 2015).” (please see last paragraph of the Statistics, thank you.
Other absolute reliability parameters, such as Standard error of measurement or Minimum detectable change, should be reported.
RESPONSE: Thank you for your suggestion. In the revised version of the manuscript SEM and MDC were calculated for intra-testing and inter-testing reliability (please see Tables 1 and 2). Also, it is now indicated in the Statistics subsection (please see previous comment where we indicated changes done in Statistics subsection according to your suggestion). Interpretation of the results for inter-testing reliability now reads: “Test-retest reliability for the newly developed PPA and NPA tests is presented in Table 1. In short, reliability of the PPA and NPA was similar, both with regard to absolute, and relative-reliability. However, the SEM of NPA was somewhat bigger (1.71 and 1.82, for PPA and NPA, respectively), which consequently generated larger MDC in NPA, than in PPA (5.03 and 2.58 for NPA and PPA, respectively). Mean testing results of the 2nd testing day were somewhat better than results obtained at 1st testing day, but with no significant differences obtained by t-test for dependent samples (t value = 0.98 and 0.91, for PPA and NPA, respectively; p > 0.05).” (please see text after Table 1). For intra-testing text now reads: “
- Results: For a better description of the sample and the improvement of external validity of the study design, please, report the sociodemographic and clinical variables of the participants. Relevant data, such as pain, time after the surgery,… are relevant.
RESPONSE: Additional details are added in Participants subsection with regard to inclusion/exclusion criteria. Also, and as specified later, validity is checked with regard to differences between age categories and gender.
- The division by age is interesting, but why 50 years? And other subdivisions according to different factors were also studied?
RESPONSE: Actually, division into 2 groups (- 50 vs. +50 yrs) was suggested in previous study of Miyamoto et al (please see References). In this version of the manuscript we tried to explain it more specifically. Also, and according to your suggestion, the differences between males and females are additionally calculated. The division is therefore explained in Statistics subsection, and text reads: “Discriminative validity of the tests was evaluated by examining the differences between specific groups of participants. In brief, previous studies showed a significant decrease in agility after 50 years of age (Miyamoto et al., 2008). Therefore, we identified the differences in NPA and PPA by comparing participants younger than 50 years and those older of 50 years. Additionally, discriminative validity was identified according to gender, by comparing males vs. females. In both cases, Student’s t-tests for independent samples were calculated for results in PPA and NPA.” (please see Statistics – text starting with Discriminative validity).
- Tables 1 and 2: The 95% confidence intervals are necessary for all outcomes.
RESPONSE: 95%ci are added for ICC (Table 1 and 2), and Means (Table 1); Thank you
- Tables 3 and 4: Again, why anthropometric variables were the only to be compared with outcomes?
RESPONSE: We must agree that other variables deserve attention with regard to studied problem, but for a moment we decided to use only anthropometrics, age and gender as possible covariates of NPA and PPA. Namely, these variables are easy to score, while testing of the NPA and PPA was time consuming, and (must say) demanding for tested participants. However, in this version of the manuscript we tried to broaden the problem and apart from correlations with anthropometrics/body built we evidenced also: (i) differences between genders, and (ii) differences between age-groups. We hope that it enriched the experimental approach and improved the applicability of our findings. Also, we paid attention on mentioning the necessity of evaluation of other variables in future studies (please see Limitation subsection)
- All Discussion and Conclusion sections should be adapted to all new perspectives.
RESPONSE: The discussions and conclusions are modified according to amendments done in statistics and results. For example; new paragraph is added in subsection 4.2 Validity (discussing the high correlation between NPA and PPA in tested subjects); further, new part of the discussion is dedicated to established differences between males and females (note that we added these analysis in the paper); the discussed issues are incorporated into Limitations and Strengths of the study; Conclusion is rewritten. For changes please see text highlighted in yellow and comments/responses bellow. Thank you.
The high correlation between both tests results could be due to a fail in construct validity. A deep discussion of this possibility could improve some parts of the text.
RESPONSE: Indeed, the correlation between NPA and PPA deserved more elaborated discussion, than it was provided in the first version of the manuscript. In this revision we tried to elaborate it in the discussion and text reads: “It must be noted that NPA and PPA were strongly correlated, and correlation between these two performances indicated that tests share more than 90% of the common variance. This is a relatively unique findings, since previous studies regularly reported relative independence of these performances, with common variance rarely exceeding 20% (Sattler et al., 2015; Sekulic, Krolo, et al., 2014; Spasic et al., 2015). However to the best of our knowledge this is the first study where different facets of agility (e.g. pre-planned and non-planned agility) were simultaneously examined in clinical population and therefore our results are comparable only with those investigations where correlations between these performances were evidenced in athletic samples. In brief, studied on athletes (e.g. trained individuals) indicated that PPA primary depends on conditioning capacities (i.e. speed, strengths, power, balance), the NPA-performance is highly influenced by perceptual, cognitive and reactive capacities of the tested participants (Scanlan et al., 2014). However, practically all studies where authors confirmed relative independence of the PPA and NPA (e.g. low correlation between PPA and NPA) included well-trained athletes, whose results on tested capacities were not variable (Pojskic et al., 2018; Scanlan et al., 2014; Sekulic et al., 2019). Meanwhile, the performances of our participants were highly variable (i.e. some participants performed 3-times better than others). Simply mathematically, such high variability of the results increased the correlation between NPA and PPA in observed sample (Huck, 2000).” Also, the problem is mentioned in the Limitation section. Text reads: “Furthermore, poor health could have resulted in a low level of agility and other reactive indicators within the agility test design. This may lead to high variability of the results, which consequently increased even the correlation between PPA and NPA in our participants.” (please see Limitations subsection).
The heterogeneity of the sample needs further and focused comments.
RESPONSE: Yes, we agree that heterogeneity of the sample is factor which deserves special attention. However, this is natural characteristic of studied sample of participants. This is specified in Limitations subsection. Text reads: “Furthermore, poor health could have resulted in a low level of agility and other reactive indicators within the agility test design. This may lead to high variability of the results, which consequently increased even the correlation between PPA and NPA in our participants. However, the high heterogeneity of the results is natural characteristic of the studied sample, and therefore can presents a “real nature” of the studied problem.”
- Line 328: A proof concept could solve these problems in the design of a new test for physical features.
RESPONSE: Actually, the pilot study specified in Methodological consideration subsection was done in order to evaluate the feasibility of concept and idea. Because of preliminary findings the original tests were shortened, and preliminary reported to be reliable. This is now indicated specifically, and text reads: “Before presenting the conclusion, we would like to acknowledge the change in testing protocols during the experiment. Initially, the pilot study was performed, where different forms of NPA and PPA testing procedures were evaluated (Perić, Spasić, & Sekulić). In brief, participants originally had to perform a test which was longer in duration (5 attempts), with a shorter additional test being added to monitor changes in the agility performance etc.” (please see subsection 4.4; thank you).
- Please, revise the references, because some details are lacking (for example: 2, 10, 23).
RESPONSE: References are revised according to Journal’s recommendations. Thank you.
Thank you once again!
Staying at your disposal
Authors
Reviewer 2 Report
Diagnostics of the Pre-Planned and Non-Planned Agility in a Clinical Population, etc.
The aim of the study was to evaluate the reliability, validity, and correlates of newly developed tests of non-planned agility (NPA) and pre-planned agility (PPA) in an untrained/clinical population.
The study was well designed, planned and implemented. Statistics is fine and convincing.
Significant differences were found between age groups (+50 years vs. -50 years of age), with a better performance being recorded in the younger group, which confirmed the validity of the NPA and PPA tests.
Conclusions that the proposed tests are reliable, valid, easy to score, quick to administer, and useful when evaluating agility characteristics in an untrained/clinical population after knee arthroplasty/arthroscopy are supported by results.
Discussion of additional publications will extend the readership and circulation of the results. Examples follows:
Sarabon, N.; Kern, H.; Loefler, S.; Jernej, R. Selection of body sway parameters according to their sensitivity and repeatability. Eur. J. Transl. Myol. 2010, 20, 5.
Figueiredo, A.I.; Balbinot, G.; Brauner, F.O.; Schiavo, A.; Baptista, R.R.; Pagnussat, A.S.; Hollands, K.; Mestriner, R.G. SPARC Metrics Provide Mobility Smoothness Assessment in Oldest-Old with and Without a History of Falls: A Case Control Study. Front. Physiol. 2020, 11, 540.
Cabral, K.d.N.; Brech, G.C.; Alonso, A.C.; Soares, A.T.; Opaleye, D.C.; Greve, J.M.D.; Jacob-Filho, W. Posturographic measures did not improve the predictive power to identify recurrent falls in community-dwelling elderly fallers. Clinics 2020, 75, e1409.
Akizuki, K.; Echizenya, Y.; Kaneno, T.; Ohashi, Y. Usefulness of an unstable board balance test to accurately identify community-dwelling elderly individuals with a history of falls. J. Rehabil. Med. 2019, 51, 71–76
Sarabon, N.; Loefler, S.; Cvecka, J.; Sedliak, M.; Kern, H. Strength training in elderly people improves static balance: A randomized controlled trial. Eur. J. Transl. Myol. 2013, 23, 85–95
Cvecka, J.; Tirpakova, V.; Sedliak, M.; Kern, H.; Mayr, W.; Hamar, D. Physical activity in elderly. Eur. J. Transl. Myol. 2015, 25, 249–252
Park, S.H. Tools for assessing fall risk in the elderly: A systematic review and meta-analysis. Aging Clin. Exp. Res. 2018, 30, 1–16.
Poe, S.S.; Dawson, P.B.; Cvach, M.; Burnett, M.; Kumble, S.; Lewis, M.; Thompson, C.B.; Hill, E.E. The Johns Hopkins Fall Risk Assessment Tool: A Study of Reliability and Validity. J. Nurs. Care Qual. 2018, 33, 10–19.
Zampieri, S.; Mosole, S.; Löfler, S.; Fruhmann, H.; Burggraf, S.; Cvečka, J.; Hamar, D.; Sedliak, M.; Tirptakova, V.; Šarabon, N.; et al. Physical exercise in Aging: Nine weeks of leg press or electrical stimulation training in 70 years old sedentary elderly people. Eur. J. Transl. Myol. 2015, 25, 237–242
Author Response
REVIEWER 2
The aim of the study was to evaluate the reliability, validity, and correlates of newly developed tests of non-planned agility (NPA) and pre-planned agility (PPA) in an untrained/clinical population.
The study was well designed, planned and implemented. Statistics is fine and convincing.
Significant differences were found between age groups (+50 years vs. -50 years of age), with a better performance being recorded in the younger group, which confirmed the validity of the NPA and PPA tests.
Conclusions that the proposed tests are reliable, valid, easy to score, quick to administer, and useful when evaluating agility characteristics in an untrained/clinical population after knee arthroplasty/arthroscopy are supported by results.
RESPONSE: Thank you for your support and for your comments. We strictly followed your suggestions and amended manuscript accordingly. Please see bellow how we incorporated it in the manuscript and where to find the specified changes.
Discussion of additional publications will extend the readership and circulation of the results. Examples follows:
Sarabon, N.; Kern, H.; Loefler, S.; Jernej, R. Selection of body sway parameters according to their sensitivity and repeatability. Eur. J. Transl. Myol. 2010, 20, 5.
Figueiredo, A.I.; Balbinot, G.; Brauner, F.O.; Schiavo, A.; Baptista, R.R.; Pagnussat, A.S.; Hollands, K.; Mestriner, R.G. SPARC Metrics Provide Mobility Smoothness Assessment in Oldest-Old with and Without a History of Falls: A Case Control Study. Front. Physiol. 2020, 11, 540.
Cabral, K.d.N.; Brech, G.C.; Alonso, A.C.; Soares, A.T.; Opaleye, D.C.; Greve, J.M.D.; Jacob-Filho, W. Posturographic measures did not improve the predictive power to identify recurrent falls in community-dwelling elderly fallers. Clinics 2020, 75, e1409.
Akizuki, K.; Echizenya, Y.; Kaneno, T.; Ohashi, Y. Usefulness of an unstable board balance test to accurately identify community-dwelling elderly individuals with a history of falls. J. Rehabil. Med. 2019, 51, 71–76
Sarabon, N.; Loefler, S.; Cvecka, J.; Sedliak, M.; Kern, H. Strength training in elderly people improves static balance: A randomized controlled trial. Eur. J. Transl. Myol. 2013, 23, 85–95
Cvecka, J.; Tirpakova, V.; Sedliak, M.; Kern, H.; Mayr, W.; Hamar, D. Physical activity in elderly. Eur. J. Transl. Myol. 2015, 25, 249–252
Park, S.H. Tools for assessing fall risk in the elderly: A systematic review and meta-analysis. Aging Clin. Exp. Res. 2018, 30, 1–16.
Poe, S.S.; Dawson, P.B.; Cvach, M.; Burnett, M.; Kumble, S.; Lewis, M.; Thompson, C.B.; Hill, E.E. The Johns Hopkins Fall Risk Assessment Tool: A Study of Reliability and Validity. J. Nurs. Care Qual. 2018, 33, 10–19.
Zampieri, S.; Mosole, S.; Löfler, S.; Fruhmann, H.; Burggraf, S.; Cvečka, J.; Hamar, D.; Sedliak, M.; Tirptakova, V.; Šarabon, N.; et al. Physical exercise in Aging: Nine weeks of leg press or electrical stimulation training in 70 years old sedentary elderly people. Eur. J. Transl. Myol. 2015, 25, 237–242
RESPONSE: Thank you for your suggestions. Most of the specified references are included in the revised version of the paper. For example,
Sarabon, N.; Kern, H.; Loefler, S.; Jernej, R. Selection of body sway parameters according to their sensitivity and repeatability. Eur. J. Transl. Myol. 2010, 20, 5.
“Specifically, it is well-established that stability requires a well-functioning muscular-skeletal system, as well as an intact balance system (Davis et al., 2004; Sarabon, Kern, Loefler, & Jernej, 2010).” (please see Introduction)
Figueiredo, A.I.; Balbinot, G.; Brauner, F.O.; Schiavo, A.; Baptista, R.R.; Pagnussat, A.S.; Hollands, K.; Mestriner, R.G. SPARC Metrics Provide Mobility Smoothness Assessment in Oldest-Old with and Without a History of Falls: A Case Control Study. Front. Physiol. 2020, 11, 540.
“Both NPA and PPA were found to be valid in the differentiation of the age groups observed, and in both cases, the younger group (<50 years of age) achieved better results than the older group (>50 years of age). Although we could not find a study which simultaneously investigated NPA and PPA in an untrained population, knowing the aging-related neuromuscular and neurocognitive declines, we may say that evidenced differences were expected (Figueiredo et al., 2020).” - DISCUSSION
Sarabon, N.; Loefler, S.; Cvecka, J.; Sedliak, M.; Kern, H. Strength training in elderly people improves static balance: A randomized controlled trial. Eur. J. Transl. Myol. 2013, 23, 85–95
“It is well-established that stability requires a well-functioning muscular-skeletal system, as well as an intact balance system (Davis et al., 2004; Nejc, Loefler, Cvecka, Sedliak, & Kern, 2013; Sarabon, Kern, Loefler, & Jernej, 2010).” – INTRODUCTION
etc.
For more details, please see list of References. Thank you!
Reviewer 3 Report
Dear Authors,
I want to thank you for a very well written paper in an area that is often under researched, as agility is not well defined in clinical populations. I have a few minor suggestions and a broad methods question. Overall a good paper that answers many questions.
Minor edits.
Can the authors please add Age to the correlations tables (2-5)? It is of importance to see not only the relationship between NPA and PPA to age but also other variables like body fat and lean body mass.
Methods question:
This was semi addressed in the limitations section but why was there four attempts per trial. This seems that the reaction part could have been assessed by a single test of reacting to the light sitting in the chair and returning to the start line. Why was five originally chosen and then reduced to four and why not one. The fatigue factor was discussed but also maybe major implications of reaction time between each additional attempt. I do not know if you have that data of when each time the infrared beam was broken but if you had this data, it would make this paper even better.
Author Response
REVIEWER 3
Dear Authors,
I want to thank you for a very well written paper in an area that is often under researched, as agility is not well defined in clinical populations. I have a few minor suggestions and a broad methods question. Overall a good paper that answers many questions.
RESPONSE: Thank you for recognizing the importance and potential of our work. Also, thank you for your comments and suggestions. Please see bellow how we responded and where to find amendments.
Minor edits.
Can the authors please add Age to the correlations tables (2-5)? It is of importance to see not only the relationship between NPA and PPA to age but also other variables like body fat and lean body mass.
RESPONSE: As you suggested, the age is added as the variable in correlation table. Please see tables 3, 4 and 5: Also, results are presented in text. Thank you.
Methods question:
This was semi addressed in the limitations section but why was there four attempts per trial. This seems that the reaction part could have been assessed by a single test of reacting to the light sitting in the chair and returning to the start line. Why was five originally chosen and then reduced to four and why not one. The fatigue factor was discussed but also maybe major implications of reaction time between each additional attempt. I do not know if you have that data of when each time the infrared beam was broken but if you had this data, it would make this paper even better.
RESPONSE: Thank you for your suggestion. Actually, when we finalized this paper we thought that it will be important to analyze “only one” attempt (first one for example – as you suggested). However, we must note that participants were “instructed” that they will have more attempts, and had familiarization trials where they “practiced” four trials. Consequently, we are positive that they didn’t perform “maximally” over first attempt. Therefore, in this paper we didn’t analyse first attempt as a final test result, but will certainly do testing where this issue will be specifically analyzed. Thanks to your comment, this issue is presented in Limitations and specified as future direction for investigation. Text reads: “In future studies this issue may be solved by evaluating tests consisting of only one attempt, either for NPA or PPA performance. One could argue that such analysis is possible even now (i.e. every attempt is recorded), but in this study participants were initially informed about number of attempts and therefore almost certainly did not execute first attempt maximally.” (please see Limtiations and strengths, thank you)
Staying at your disposal
Authors
Round 2
Reviewer 1 Report
Unfortunately, the study has fatal flaws. Revising it will not fix it.
Examples:
- The change in the original objectives cannot solve all specific concerns. Further, the new limitations of the study are insufficient to keep the clinical relevance of the study, since other variables, such as balance, quality of life, pain,…, are part of the clinical interest of the agility tests.
“What do the authors mean with: certain correlates? In light of the Results section, many other variables could increase the relevance of the results, such as balance, quality of life, pain,…
RESPONSE: Indeed, the studied correlates were specified explained in the original version of the paper. Therefore, the part of the text where aims of the study is presented is rewritten and now reads: “Therefore, the aim of this study was to evaluate the reliability, validity, and usefulness of newly developed NPA and PPA tests among subjects involved in the rehabilitation process after knee arthroplasty and knee arthroscopy (clinical population). Additionally, we evaluated anthropometric indices, age and gender as factors associated with NPA and PPA in studies participants.” (please see end of the last paragraph of the Introduction section). Also, we totally agree with your opinion that other variables deserve attention too, and this is now specifically stated in the Limitations of the study. Text reads: Finally, this study evaluated only small number of correlates of PPA and NPA Also, further studies should evaluate other variables than those studies in this investigation as being potentially correlated to PPA and NPA in clinical population.” (please see 1st paragraph of the Limitations and strengths subheading – before Conclusion)”
- Too late to change inclusion/exclusion criteria.
“Line 96: The exclusion criteria should be better detailed. Did not the authors exclude vestibular diseases? And previous surgeries in the low limbs? And what about drugs?
RESPONSE: Thank you for your suggestion. In the revised version of the manuscript more details are added about exclusion criteria. Text reads: Exclusion criteria included inability to walk without knee pain on a distance of 100 meters (10x10 meters), pain while sitting down and standing up from the chair, pain and short of breath during 5 min walk in normal pace. Further, participants were not included in the investigation if they had balance disorder while standing/walking (irrespective of accompanied wooziness, and/or sensation of movement, spinning, floating), or if they consumed drugs that could be used in treatment of balance disorders (i.e. aminoglycoside antibiotics, Beta-histine).” (please see subsection on Participants).”
- Too late to include an “a priori” sample size.
“An a priori sample size calculation is lacking. Anyway, a power analysis of the results is also lacking.
RESPONSE: Indeed, in the original version of the manuscript we missed to report that the required sample was calculated on the basis of preliminary pilot study. It is now specified, and text reads: On the basis of results from preliminary pilot study (Perić, Spasić, & Sekulić, 2019), statistical power of 0.8, and significance level of 0.05, the required sample of participants was 34.” (Please see last sentence of Participants subsection)”
- The inclusion of Age is insufficient to increase the relevance of the Results.
“Tables 3 and 4: Again, why anthropometric variables were the only to be compared with outcomes?
RESPONSE: We must agree that other variables deserve attention with regard to studied problem, but for a moment we decided to use only anthropometrics, age and gender as possible covariates of NPA and PPA. Namely, these variables are easy to score, while testing of the NPA and PPA was time consuming, and (must say) demanding for tested participants. However, in this version of the manuscript we tried to broaden the problem and apart from correlations with anthropometrics/body built we evidenced also: (i) differences between genders, and (ii) differences between age-groups. We hope that it enriched the experimental approach and improved the applicability of our findings. Also, we paid attention on mentioning the necessity of evaluation of other variables in future studies (please see Limitation subsection)”